# Runtime Attestation for Secure LLM Serving in Cloud-Native Trusted Execution Environments

Jianchang Su, Wei Zhang*

University of Connecticut,

*Corresponding author

*jianchang.su@uconn.edu, wei.13.zhang@uconn.edu*

*Abstract*—Serving Large Language Models (LLMs) in cloud environments introduces significant security challenges, particularly protecting sensitive data from untrusted cloud components. While Trusted Execution Environments (TEEs) provide hardware isolation, current approaches offer only boot-time attestation without container orchestration integration. We present a framework addressing these limitations through: runtime attestation for continuous integrity verification, container-level measurement for multi-tenant environments, attestation-aware Kubernetes integration, and hardware-agnostic TEE abstraction. This comprehensive approach creates an unbroken chain of trust from hardware to application, enabling secure LLM deployment against both infrastructure and orchestration-level attacks while maintaining cross-platform compatibility.

*Index Terms*—Trusted Execution Environment, Runtime Attestation, Large Language Models, Confidential Computing

## I. INTRODUCTION

The rapid growth of large-language models (LLMs) and their increasing adoption in real-world applications has led to a paradigm shift in how natural-language-processing (NLP) services are developed and deployed [8]. Cloud-native architectures, characterized by containerized micro-services orchestrated by platforms like Kubernetes, have emerged as the dominant approach for scalable and resilient LLM serving [24], [34]. However, the deployment of LLMs in cloud-native environments introduces significant security challenges, particularly in protecting the confidentiality and integrity of the model weights, user queries, and generated responses [19].

Confidential computing technologies, such as Intel Trust Domain Extensions (TDX) [22], offer a promising solution to these challenges by providing hardware-based isolation and protection for sensitive workloads. However, existing confidential-computing solutions are primarily designed for monolithic, boot-time–attested, VM-centric deployments and lack native integration with the dynamic, microservice-oriented nature of cloud-native LLM serving [5]. Moreover, the heterogeneity of hardware-based Trusted Execution Environments (TEEs) across different cloud providers and platforms creates vendor lock-in risks and hinders the portability of secure LLM services [30].

To address these challenges, we propose a comprehensive system for enabling runtime attestation and secure execution of LLM workloads in confidential cloud-native environments. Our approach extends the capabilities of Intel TDX to support continuous integrity monitoring and attestation of the underlying infrastructure, seamlessly integrates this functionality with cloud-native orchestration frameworks, and introduces a hardware-agnostic TEE runtime for securing individual LLM serving components. The proposed system comprises four key contributions:

- A framework for runtime attestation in Intel TDX-based confidential containers, leveraging hardware security features and kernel-level extensions to establish an unbroken chain of trust from the container's launch to its runtime execution.
- A set of enhancements to cloud-native orchestration platforms like Kubernetes to enable pod-level attestation and secure scheduling of confidential LLM serving workloads.
- A hardware-agnostic TEE runtime that provides a unified interface for secure execution and data protection within LLM serving microservices, abstracting the complexities of different TEE technologies.
- A TEE-aware LLM serving scheduler that intelligently places and scales LLM workloads based on the attested security state of the underlying infrastructure, optimizing for both security and performance.

Our system unifies hardware security, cloud orchestration, and LLM-specific needs to enable secure, efficient cloud deployment of language models-supporting trusted, scalable use in applications like chatbots, content moderation, and sentiment analysis.

## II. BACKGROUND AND MOTIVATION

### A. Background

**LLM Serving in Cloud-Native Environments.** The rapid adoption of LLMs in various applications has led to a surge in demand for efficient and scalable LLM serving infrastructures [1], [3], [8], [23], [24]. Cloud-native architectures, characterized by containerized microservices orchestrated by platforms like Kubernetes, have emerged as the dominant paradigm for deploying and managing LLM workloads [3], [23], [24], [34]. These architectures enable automatic scaling, fault tolerance, and resource efficiency, making them well-suited for the dynamic and resource-intensive nature of LLM serving.

However, the deployment of LLMs in cloud-native environments also introduces new security challenges. The distributed

TABLE I: Feature Support in TEE Platforms and Frameworks ($\checkmark$ = supported, $\circ$ = partially supported, $\times$ = not supported)

| Platform/Framework | Runtime Attest. | Container Security | K8s Integ. | HW Agnostic |
|---|---|---|---|---|
| Intel SGX [21] | $\checkmark$ | $\times$ | $\circ$ | $\times$ |
| Intel TDX | $\circ$ | $\circ$ | $\circ$ | $\times$ |
| AMD SEV/SEV-SNP [2] | $\circ$ | $\circ$ | $\circ$ | $\times$ |
| ARM TrustZone/CCA [4] | $\circ$ | $\circ$ | $\times$ | $\times$ |
| Confid. Containers [11] | $\circ$ | $\checkmark$ | $\checkmark$ | $\times$ |
| Azure Confid. Cont. [27] | $\circ$ | $\checkmark$ | $\checkmark$ | $\times$ |
| **Proposed System** | $\checkmark$ | $\checkmark$ | $\checkmark$ | $\checkmark$ |

and ephemeral nature of containers, coupled with the reliance on potentially untrusted infrastructure, exposes LLM workloads to various threats, such as data breaches, insider attacks, and malicious interference [9], [16], [19]. Protecting the confidentiality and integrity of the model weights, user queries, and generated responses throughout the serving lifecycle is crucial for building trustworthy and reliable LLM services.

**Confidential Computing and Intel TDX.** Confidential computing technologies aim to protect sensitive workloads from unauthorized access or tampering, even in untrusted environments. TDX [22] is a prominent example of such technologies, providing hardware-based isolation and encryption for virtual machines (VMs). TDX introduces the concept of a "trust domain" - a hardware-enforced security boundary that shields the VM's memory from the host and other VMs.

However, TDX in its current implementation only supports attestation at boot time (launch-time attestation) [10]. This critical limitation means that while TDX can verify the integrity of a VM when it initializes, it cannot guarantee runtime security after the VM has booted. Consequently, cloud service providers, potential adversaries, or compromised components can still access VM memory or tamper with running workloads through various attack vectors, such as unauthorized SSH access, malicious kernel modules, or exploitation of CPU interfaces [31], [36]. Extending TDX with runtime attestation capabilities is crucial for ensuring the continuous protection of sensitive workloads throughout their lifecycle [32].

### B. Motivation

Despite significant advances in confidential computing technologies, our analysis reveals critical gaps in the existing infrastructure for secure LLM serving in cloud-native environments. Current solutions provide only partial protection throughout the LLM lifecycle, particularly in dynamic containerized deployments. To systematically identify these gaps, we conducted a comprehensive analysis of major hardware TEE platforms and relevant research frameworks. Table 1 presents our comparative findings, highlighting the limitations of existing approaches against the four key features required for secure cloud-native LLM serving.

As shown in the Table I, existing solutions have significant limitations across multiple dimensions. Runtime attestation capabilities are predominantly limited to launch-time verification, with Intel SGX offering strong initial code attestation but lacking continuous monitoring capabilities. Container-level security varies widely, with process-based TEEs like SGX requiring complex LibOS solutions while VM-based TEEs need additional guest mechanisms to protect individual containers. Kubernetes integration remains superficial across platforms, focusing primarily on resource allocation or runtime selection without incorporating attestation results into scheduling decisions. The hardware heterogeneity across TEE technologies creates significant portability challenges, with existing abstraction efforts remaining either conceptual or limited in scope.

These limitations are particularly problematic for LLM serving applications, where the sensitivity of model weights, user queries, and generated responses demands comprehensive protection throughout the execution lifecycle [7]. The dynamic nature of cloud-native LLM serving further complicates these challenges, requiring solutions that can verify workload integrity while maintaining compatibility with orchestration frameworks [37].

Our proposed system addresses these gaps by introducing a comprehensive solution with four integrated components: a continuous runtime attestation framework, enhanced container-level protection, native Kubernetes integration for pod-level attestation, and a hardware-agnostic TEE runtime. This approach enables a new level of trust and efficiency for secure LLM services in cloud-native environments.

## III. THREAT MODEL

To secure LLM deployments in cloud-native TEE environments, we define a threat model addressing the unique convergence of LLMs, confidential computing, and container orchestration.

### A. Assets and Adversaries

Our system protects several critical assets: (1) LLM model intellectual property including architecture and parameters [15], (2) sensitive user prompts and responses [19], (3) dynamic runtime state including memory and conversation context [18], and (4) security-critical system configurations [6].

We consider four primary adversary types:

- **Cloud Infrastructure Adversary:** Controls host OS and hypervisor, can observe network traffic, inspect unprotected memory, and attempt side-channel attacks [33].
- **Orchestration Plane Adversary:** Controls Kubernetes components and can manipulate pod specifications, configurations, and scheduling decisions [14].
- **Co-tenant Adversary:** Malicious workloads on the same physical host capable of mounting side-channel attacks [32].
- **Malicious User/Compromised Application:** External users crafting adversarial inputs or compromised internal components [39].

### B. Threat Vectors and Mitigation

We identify three primary threat categories that current solutions address inadequately:

**Runtime Integrity Violations:** While TEEs like Intel TDX provide strong boot-time attestation, they cannot detect compromises occurring after initialization [38]. A malicious orchestrator or exploited vulnerability could inject code post-boot, compromising the TEE's integrity while existing attestation mechanisms remain oblivious. Our continuous runtime attestation addresses this gap by providing ongoing verification throughout the workload lifecycle.

**Orchestration-Level Attacks:** Standard TEE protections focus on hardware isolation but neglect the powerful Kubernetes control plane that manages TEE deployments [35]. An adversary could schedule confidential workloads onto non-TEE nodes, inject malicious configurations, or tamper with attestation workflows. Our deep Kubernetes integration with pod-level attestation and security-aware scheduling provides protection against these orchestration-layer threats.

**Hardware Vendor Lock-in:** Current solutions tie deployments to specific TEE implementations, creating security and portability risks [28]. Our hardware-agnostic TEE runtime provides a unified interface across diverse TEE technologies, improving resilience against vendor-specific vulnerabilities while simplifying deployment across heterogeneous infrastructure.

### C. Trust Assumptions

Our system relies on a minimal Trusted Computing Base (TCB) including the Intel CPU with TDX support, the TDX Module, Memory Encryption Engine, and the initial TD software stack verified through attestation [10]. All components outside this TCB (host OS, hypervisor, Kubernetes control plane, network, and storage) are considered untrusted [13].

We explicitly consider sophisticated physical attacks requiring specialized equipment, hardware supply chain compromises, pure denial-of-service attacks, and inherent LLM algorithmic vulnerabilities outside our scope of protection [17].

This threat model directly motivates our system design, demonstrating how each component addresses specific gaps in current confidential computing approaches for cloud-native LLM serving.

### IV. System Design

To address the identified limitations in secure cloud-native LLM serving, we present a comprehensive system that enables continuous attestation and secure execution of LLM workloads in TDX-based environments shown in Figure 1.

### A. Runtime Attestation for TDX

The foundation of our system extends Intel TDX to support continuous integrity measurement during VM execution. We introduce an RTMR extension mechanism that creates an unbroken chain of trust from boot time to runtime, enabling the guest OS to securely update Runtime Measurement Registers during execution. Our implementation includes a secure interface between the guest OS and TDX module, with robust access control and a standardized measurement protocol that supports cryptographic authentication.

We integrate this mechanism with the Linux Integrity Measurement Architecture (IMA), allowing IMA to use TDX's

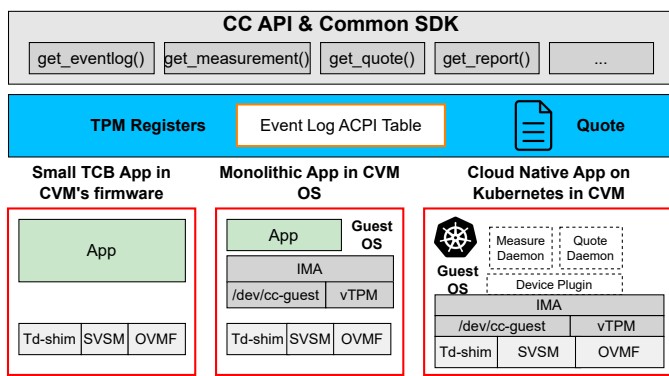

Fig. 1: Proposed system Architecture

RTMRs as its trust anchor. This integration provides fine-grained visibility into the VM's runtime state, capturing critical events such as kernel module loading and configuration changes. By enabling continuous verification, this component addresses the post-boot compromise detection gap identified in our threat model.

### B. Confidential Integrity Measurement Agent

The Confidential Integrity Measurement Agent (CIMA) extends runtime attestation capabilities to the container level. By making IMA container-aware through Linux cgroups, CIMA enables separate integrity measurements for each container within a confidential VM. We implement integrations with container runtimes to measure images during their lifecycle, calculating cryptographic digests of layers and extending these measurements into the RTMRs.

CIMA includes a gRPC-based service that exposes an attestation API to containers, allowing them to request integrity measurements and attestation reports. To simplify adoption, we provide language-specific SDKs that abstract the attestation process with intuitive interfaces. This component enables verification of individual microservices within shared confidential environments, addressing the container-level security challenges identified in our analysis.

### C. Kubernetes Attestation Integration

To leverage attestation capabilities in orchestration decisions, we integrate with Kubernetes through several extensions. Our custom Admission Webhook verifies container image integrity before deployment by comparing measurements against expected values from a trusted repository. The TEE-aware Scheduler Plugin considers attestation state when placing workloads, ensuring sensitive applications run only on verified nodes.

We develop an Attestation Policy Engine that enables administrators to define attestation-based security policies as Kubernetes resources, controlling deployment based on trust verification. This component addresses orchestration-level attack vectors by ensuring Kubernetes scheduling and policy decisions incorporate verified trust information.

TABLE II: Confidential Computing API Core Functions

| API Function | Description |
|---|---|
| `get_default_algo()` | Returns default cryptographic algorithms supported by the TEE |
| `get_measure_count()` | Gets number of available measurement registers |
| `get_measurement(imr)` | Retrieves specific measurement register value |
| `get_quote(nonce, data)` | Generates attestation report with replay protection |
| `get_eventlog(start, count)` | Retrieves portion of TEE's integrity event log |

### D. Confidential Computing API

The Confidential Computing API provides a hardware-agnostic abstraction layer for confidential computing operations shows in Table II. This API enables portable, vendor-agnostic application development across diverse TEE technologies.

We implement adapters for multiple TEE technologies, including Intel TDX, Intel SGX, and AMD SEV-SNP. Each adapter translates abstract API calls into technology-specific operations, handling different attestation protocols and security models. This design enables applications to switch between TEE technologies by changing the adapter configuration, addressing hardware vendor lock-in issues identified in our threat model.

Together, these four components provide a comprehensive solution for secure LLM serving in cloud-native environments, enabling continuous integrity verification, container-level security, security-aware orchestration, and hardware-agnostic deployment of confidential workloads.

## V. LIMITATIONS & FUTURE WORK

**Security Analysis Limitations:** While our system provides comprehensive protection against the threat vectors identified in our model, several limitations remain. Our current approach does not fully mitigate side-channel attacks, which can potentially leak sensitive information through microarchitectural effects. These attacks remain challenging for all TEE solutions, as they exploit hardware behaviors outside the TEE's control. Additionally, our threat model assumes the correctness of the hardware TCB, which may not hold if hardware vulnerabilities are discovered in the underlying TDX implementation.

**Performance Overhead:** The runtime attestation mechanisms introduce computational overhead, particularly for container-level integrity measurement. While our preliminary evaluations show acceptable performance for LLM serving workloads, the overhead increases with the number of containers and the frequency of attestation. Future optimization work is needed to reduce this overhead, especially for latency-sensitive LLM inference scenarios that require high throughput and real-time responses.

**Attestation Freshness Guarantees:** Our current system provides periodic attestation, which creates a potential time window between attestations during which malicious changes might go undetected. Although we minimize this window through careful timing, achieving true continuous attestation remains challenging. Future work could explore event-driven attestation triggers and more efficient incremental measurement techniques to further reduce this gap.

## VI. RELATED WORK

**Trusted Execution for LLM Security.** Recent advancements in LLM security have explored various TEE-based protection mechanisms. Li et al. [20] and Gim et al. [19] demonstrated how TEEs can protect sensitive model components and user prompts in distributed LLM environments through hardware-enforced isolation and lightweight encryption. These approaches build upon foundational TEE research by Li et al. [26], who established a comprehensive taxonomy for TEE-based secure computation protocols and evaluation criteria for comparing security approaches. However, Muñoz and Ríos [29] identified critical vulnerabilities in existing TEE implementations, highlighting that current solutions require additional safeguards beyond hardware isolation. A significant limitation in current research is the predominant focus on static attestation at launch time rather than continuous runtime verification [10], [13]. This critical gap leaves systems vulnerable to runtime attacks after initial attestation, particularly in long-running LLM serving environments. Our work directly addresses this limitation by developing a runtime attestation framework specifically designed for LLM serving in cloud-native environments.

**Container-Level Attestation in Cloud Environments.** As containerization has become the standard deployment model for cloud applications, securing containerized workloads in TEEs has emerged as a critical research direction. The CNCF Confidential Containers project [6] and commercial implementations from Microsoft [25] and Alibaba [12] provide hardware-enforced isolation for containerized applications using various TEE technologies (AMD SEV, Intel TDX, Intel SGX). However, these solutions focus primarily on data-in-use protection rather than comprehensive attestation frameworks and lack deep integration with orchestration systems. Thijsman et al. [35] identified critical gaps in existing trusted cloud-native deployments, highlighting the need for attestation-aware orchestration but without addressing the container-level granularity required for multi-tenant LLM serving. Our research extends these efforts by developing a comprehensive framework that integrates runtime attestation with Kubernetes orchestration, enabling secure LLM serving with strong protection against both infrastructure and orchestration-level attacks while supporting hardware-agnostic deployment across diverse TEE technologies.

## VII. CONCLUSION

We presented a system addressing critical security gaps in confidential computing for LLM deployments through continuous runtime attestation, container-level integrity verification, and security-aware Kubernetes orchestration. While empirical evaluation remains future work, we believe our approach

will significantly enhance LLM security posture against both infrastructure and orchestration-level attacks through post-attestation verification and hardware-agnostic deployment. Future research will explore performance optimization, specialized LLM security features, and secure distributed inference across multiple confidential nodes.

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
