# OpenReview forum: "Runtime Attestation for Secure LLM Serving in Cloud-Native Trusted Execution Environments"
_iscaconf.org/ISCA/2025/Workshop/MLArchSys — MLArchSys 2025 Oral_

### Official Review · Reviewer_XxtC · 2025-05-18
**Review of Runtime Attestation for Secure LLM Serving in Cloud-Native Trusted Execution Environments**

**Confidence:** 1
**Rating:** 4

**Detailed Feedback And Questions For Authors:**

Paper proposes a system for Confidential LLM serving deployments with the following guarantees: Continuous integrity verification, Container-level security, Security-aware orchestration and Hardware-agnostic deployment. The paper does a good enough job in defining a right threat model and designing a system to meet these requirements.

There are some cons to this work that limit the work from reaching full potential to be a complete paper:
- While the claim is that the system is hardware agnostic, one of the components of the system (Confidential Computing API) requires hardware specific implementations and lowering to technology specific operations.
- Reverse engineering the timing between attestations may allow an attacker to ingest attacks at the right timing.
- Some aspects in the future work section: "Security Analysis implications" & "Performance overhead" need to be addressed to some detail for this work to be considered complete.

**Top Reasons To Accept The Paper:**

Paper proposes a system for Confidential LLM serving deployments with the following guarantees:
- Continuous integrity verification
- Container-level security
- Security-aware orchestration
- Hardware-agnostic deployment

The paper does a good enough job in defining a right threat model and designing a system to meet these requirements.

**Top Reasons To Reject The Paper:**

- Relevance of the paper topic to the subject area of the MLArchSys is not clear. While MLArchSys does cover "Generative AI for security and vulnerability detection", it does not include security of the Execution environment for running LLms
- The ideas and techniques presented in the paper to achieve the 4 goals (continuous integrity verification, container-level security, security-aware orchestration, and hardware-agnostic deployment of confidential workloads) are too high level, and no details of the implementation are specifically discussed. This is a very shallow description of high level APIs or system structure.
- There are no results presented in the paper on the following and I think a proposal of a system architecture like this is incomplete without more insight into these factors.
  - Performance or Functional implications of these techniques when implemented in a real world system
  - How does the system provide comprehensive protection against the threat vectors identified in the model?
- While the claim is that the system is hardware agnostic, one of the components of the system (Confidential Computing API) requires hardware specific implementations and lowering to technology specific operations.

---

### Official Review · Reviewer_prFn · 2025-05-18
**Overall, the paper tackles a very important topic. While the paper is well written, it does not provide much technical depth of the proposed solution or more importantly any evaluation of performance impact.**

**Confidence:** 1
**Rating:** 4

**Detailed Feedback And Questions For Authors:**

Overall, I think the paper is tackling a very important issue. At a high-level the paper is also well written.

The major limitation of the current submission is the lack of any empirical evaluation of the proposed system. Without performance data and a practical demonstration of the system, it's difficult to assess the real-world viability, overhead, and scalability of the proposed framework.

Perhaps a minor comment is that the paper is a bit had to read for non-experts in the field (it assumes a certain level of familiarity with Kubernetes for example).

**Top Reasons To Accept The Paper:**

-- Tackling a very important topic -- securing LLM serving in cloud environments

-- Well written paper with clear description of threat model and categorization of proposed solution/system

**Top Reasons To Reject The Paper:**

I think the major limitation of the paper is the lack of empirical evaluation of the proposed solution.

It is unclear what are the implications on performance (latency, throughput) off LLM serving with the proposed system.

Without empirical evidence, it is hard to substantiate the claims and feasibility of the proposed solution.,

---

### Official Review · Reviewer_M1rX · 2025-05-18
**Continuous runtime and container-level attestation solution in cloud TEE, without empirical evaluation.**

**Confidence:** 4
**Rating:** 6

**Detailed Feedback And Questions For Authors:**

The paper proposes a comprehensive system to enhance the security of Large Language Model (LLM) serving in cloud-native Trusted Execution Environment (TEE) environments, specifically addressing the limitations of existing solutions which primarily offer only boot-time attestation and lack integration with dynamic container orchestration.
The core of their approach involves extending Intel TDX capabilities through a Runtime Measurement Register (RTMR) extension mechanism that allows the Guest OS's Linux Integrity Measurement Architecture (IMA) to securely update measurements during execution, thereby establishing an unbroken chain of trust from the container’s launch to its runtime execution. This enables continuous integrity verification for the Guest OS and, via a Confidential Integrity Measurement Agent (CIMA), extends this capability to individual containers within the confidential VM. Furthermore, the system integrates this attestation capability with Kubernetes for attestation-aware scheduling and policy enforcement and introduces a hardware-agnostic TEE runtime to provide a unified interface across different TEE technologies, aiming to improve security against infrastructure and orchestration-level attacks while maintaining cross-platform compatibility and addressing vendor lock-in.

Questions for authors:

*   Given that the paper states **empirical evaluation remains future work** and that preliminary evaluations show **performance overhead**, could you provide more specific details on the **expected performance impact** of the runtime attestation and container-level measurement mechanisms, particularly under varying LLM serving workloads and attestation frequencies? What specific metrics do you plan to use for evaluation?
*   The paper introduces a **Runtime Measurement Register (RTMR) extension mechanism** within Intel TDX and integrates it with **Linux IMA**. Could you elaborate on the technical details of the secure interface between the Guest OS and the TDX module for RTMR updates? How is the integrity and authenticity of the measurements ensured during this extension process, and how is access control enforced to prevent malicious updates?
*   The system provides **periodic attestation**, acknowledging a potential time window vulnerability. What is the expected or typical **minimum interval** between attestations the system can achieve? Have you considered or planned specific strategies for mitigating the risk posed by malicious changes occurring within this window?
*   The paper explicitly mentions that the current approach **does not fully mitigate side-channel attacks**. While acknowledging this is a general challenge for TEEs, have you considered any design choices or future work specifically aimed at *reducing* the attack surface or impact of *certain types* of side-channel attacks within your proposed framework?

**Top Reasons To Accept The Paper:**

1) The paper directly addresses the limitations of current Trusted Execution Environment (TEE) which primarily offers boot-time attestation, leaving the system vulnerable after initialization. The paper porposes leveraging RTMR (Runtime measurement registers) as a trust anchor for linux IMA.
2) While the proposed solution is **not specific to LLM serving**, it enables secure cloud-native deployment environment for LLMs.
3) The proposed solution is fine-grained container level security, allows Guest OS's IMA to be aware of the containers, allows separate integrity measurement for each container.
4) The paper promotes portability and reduces the vendor-specific solutions.

**Top Reasons To Reject The Paper:**

1) The paper postpones all the empirical evaluation to the future work. While they mention the potential performance overhead, no specific data is provided.
2) There are acknoweldeged security limitations such as lack of protection against different forms of side-channel attack.
3) Even though the paper proposes **continuous runtime attestation**, the time between the freshness window remains the potential risk for malicious activities. It's not clear from the paper that what is a reasonable time window for protecting the runtime environment, without significant performance overhead.

---

### Official Review · Reviewer_W5Nq · 2025-05-19
**Good proposal for cloud native micro services but not entirely related to LLM serving.**

**Confidence:** 2
**Rating:** 5

**Detailed Feedback And Questions For Authors:**

Thank you for submitting to MLArchSys. The problem of securely serving LLMs in cloud native environments is very important and relevant. I am not entirely familiar with this space. So, I enjoyed reading the paper since it provided good background and motivation sections. The paper identifies three major threat categories that is inadequately addressed by solutions today. Then it goes on to propose solutions to address each of these. I hesitate to champion this paper because of some performance concerns and concerns that it may not be fully addressing unique challenges of serving LLM models confidentially. For example, reference [20] in the paper cites a paper that addresses unique problems posed by secure distributed training of LLMs. Such an exercise of dissecting the LLM serving workload is worthwhile for this paper.

If the proposal can further identify and address unique confidential service deployment challenges unique to LLM serving, it could be a powerful proposal. Some examples of unique challenges are protecting model parameters loading, securing the query data, etc.

[20]  W. Huang, Y. Wang, A. Cheng, A. Zhou, C. Yu, and L. Wang, “A fast, performant, secure distributed training framework for large language model".

**Top Reasons To Accept The Paper:**

This paper is well written. The motivation for the problem is relevant because LLM serving could be deployed in a cloud native micro services architecture. The orchestration-layer threats are recognized challenges in deploying confidential computing workloads at scale.

**Top Reasons To Reject The Paper:**

The proposed solution, although promising, seems to have some limitations. Primarily, performance overheads could be significant and non-determinable with gRPC based attestation services. This is especially important because  LLM serving is a delay sensitive workload.